# The Genus *Dacryodes* Vahl.: Ethnobotany, Phytochemistry and Biological Activities

**DOI:** 10.3390/ph16050775

**Published:** 2023-05-22

**Authors:** Leseho Swana, Bienvenu Tsakem, Jacqueline V. Tembu, Rémy B. Teponno, Joy T. Folahan, Jarmo-Charles Kalinski, Alexandros Polyzois, Guy Kamatou, Louis P. Sandjo, Jean Christopher Chamcheu, Xavier Siwe-Noundou

**Affiliations:** 1Department of Pharmaceutical Sciences, School of Pharmacy, Sefako Makgatho Health Sciences University, P.O. Box 218, Pretoria 0208, South Africa; 2Department of Chemistry, Faculty of Science, University of Dschang, Dschang P.O. Box 67, Cameroon; 3Department of Chemistry, Tshwane University of Technology, Private Bag X680, Pretoria 0001, South Africa; 4School of Basic Pharmaceutical and Toxicological Sciences, College of Pharmacy, University of Louisiana at Monroe, Monroe, LA 71209, USA; 5Department of Biochemistry and Microbiology, Rhodes University, Makhanda 6140, South Africa; 6Department of Pharmaceutical Sciences, Faculty of Science, Tshwane University of Technology, Private Bag X680, Pretoria 0001, South Africa; 7Department of Chemistry, Federal University of Santa Catarina, Florianópolis 88040-900, Brazil

**Keywords:** *Dacryodes*, *Dacryodes edulis*, medicinal plant, phytochemistry, tropical diseases, bioactivity

## Abstract

*Dacryodes* Vahl. species, belonging to the Burseraceae family, are widely used in traditional medicine in tropical regions to treat a range of ailments including malaria, wounds, tonsillitis, and ringworms. This review discusses the distribution, ethnobotanical uses, phytochemistry, and bioactivities of *Dacryodes* species. The intent is to spur future research into isolating and identifying key active principles, secondary metabolites, and crude extracts, and evaluating their pharmacological and toxicological effects, as well as the mechanism of actions to understand their medicinal benefits. A systematic review of scientific electronic databases from 1963 to 2022 including Scifinder, Scopus, Pubmed, Springer Link, ResearchGate, Ethnobotany Research and Applications, Google Scholar, and ScienceDirect was conducted with a focus on *Dacryodes edulis* (G.Don) H.J. Lam and *Dacryodes rostrata* (Blume) H.J. Lam. Pharmacological data revealed that *D. edulis* isolates contain secondary metabolites and other phytochemical groups belonging to the terpenoids class with anti-microbial, anticancer, antidiabetic, antiinflammatory and hepatoprotective activities, highlighting its pharmacological potential in the therapy or management of diverse cancers, cardiovascular, and neurological diseases. Thus, phytochemicals and standardized extracts from *D. edulis* could offer safer and cost-effective chemopreventive and chemotherapeutic health benefits/regimen, or as alternative therapeutic remedy for several human diseases. Nevertheless, the therapeutic potential of most of the plants in the genus have not been exhaustively explored with regard to phytochemistry and pharmacology, but mostly complementary approaches lacking rigorous, scientific research-based knowledge. Therefore, the therapeutic potentials of the *Dacryodes* genus remain largely untapped, and comprehensive research is necessary to fully harness their medicinal properties.

## 1. Introduction

Plants have been used for a variety of purposes for thousands of years, including medicinal purposes. The use of plants for health and healing can be traced back to ancient civilizations, such as Sumeria, Mesopotamia, Egypt, Greece, and Islam. The oldest written evidence for the use of plants for health dates back to around 5000 years ago and was discovered on a Sumerian clay slab from Nagpur [1,2]. Traditional medicine encompasses a wide range of knowledge, practices, and abilities that have been developed and used by different societies over time for health maintenance, as well as the diagnosis, treatment, and prevention of physical and mental disorders [3]. The use of medicinal plants is a central aspect of traditional medicine in many cultures, and it involves the identification of useful plants through trial-and-error methods, followed by the refinement of their use through many generations. Medicinal plants produce a vast array of secondary metabolites, which are chemical compounds that are not essential for the plant’s growth and development but have potential therapeutic properties [3,4,5]. Such secondary metabolites have been used as building blocks for many pharmaceutical drugs and herbal remedies [6]. Vascular plants are estimated to comprise around 500,000 species. However, only about 10% of these species are presently utilized as medicinal plants [3,6]. Plant-based natural remedies can be produced from a diverse range of source materials, including fruits, vegetables, spices (such as cinnamon, ginger, and black pepper), leaves, bark, roots, and essential oils (including those derived from Eucalyptus, spearmint, clove, peppermint, pomegranate, olive, and grapefruit) [7,8,9,10,11,12,13,14]. 

*Dacryodes* Vahl., a genus belonging to the family Burseraceae, comprises around 40 species of trees and shrubs that are naturally found in tropical areas of Africa and the Americas. The African pear or “Safou”, scientifically known as *Dacryodes edulis* (G.Don) H.J. Lam, is the most popular species in this genus [12]. Its fruit is widely consumed in various African regions, and is used in the preparation of soups, sauces, and stews. The fruit is a rich source of protein, fat, and carbohydrates, and is considered an essential nutritional source by many. Additionally, other species in the *Dacryodes* genus are utilized for their timber, medicinal, and essential oil properties [13]. Several species of *Dacryodes* are also used in traditional medicine for treating ailments, such as fever, malaria, and skin infections. Previous studies have investigated *D. edulis*, a species of this genus, and some of these studies were covered in review reports [12,13]. In this review, we provide a comprehensive overview that focuses on the ethnobotanical uses, pharmacological, and chemical studies of plants in the *Dacryodes* genus. We systematically sourced, reviewed, discussed, and analyzed data from scientific electronic databases such as Scifinder, Scopus, Pubmed, Springer Link, ResearchGate, Ethnobotany Research and Applications, Google Scholar, and ScienceDirect from 1963 to 2022.

## 2. Literature Survey Databases

The scientific electronic databases including Scifinder, Scopus, Pubmed, Springer Link, ResearchGate, Ethnobotany Research and Applications, Google Scholar, and ScienceDirect from the years 1963–2022 were systematically searched, reviewed, discussed, and analyzed to obtain the data reported in this review. Keywords, such as Burseraceae, *Dacryodes*, and traditional use, were utilized during the search. Additional information on the botany, geographical distribution, ethnomedicinal uses, phytochemistry, and pharmacological properties of *Dacryodes* species was collected from journal articles, book chapters, books, and encyclopedias. The gathered information was critically analyzed to generate new insights and identify potential knowledge gaps for future research on *Dacryodes* species.

## 3. Description of *Dacryodes* Vahl. Species and Ethnobotany

The tropical humid woods of Africa, Southeast Asia, and America are habitat to more than 70 different *Dacryodes* species that are perennial, and evergreen trees belonging to the Burseraceae family that produce fleshy fruits [15]. The genus name originates from the Greek word “Dakrun,” which means “tear (drop),” referring to the resin droplets that exude from the tree’s bark. The species in this genus typically grow into small to medium-sized trees [16]. *Dacryodes edulis* (G.Don) H.J. Lam, *Dacryodes rostrata* (Blume) H.J. Lam, *Dacryodes buettneri* (Engel.) H.J. Lam, and *Dacryodes klaineana* (Pierre) H.J. Lam are the most reported species. However, information on most of the species in this genus is still scarce. Despite being undervalued, various *Dacryodes* species are used in ethnomedicine to treat various conditions, including headaches, malaria, skin conditions, fever, and anemia [15]. Moreover, the *Dacryodes* species are highly valued and prized for their timber properties, and oil-rich, delicious flavorful fruits (Figure 1) [15]. *Dacryodes* species have a long history of medical efficacy, and Native Americans have traditionally used them extensively as ethnomedicine to treat various illnesses. For example, *D. edulis* fruits, bark, leaves, resin, and other parts have been utilized to treat malaria, fever, and headaches, but data on most of the species in this genus are still scarce [16,17]. The stem bark, roots, resin, and leaves of *D. edulis* are used to treat fever, tonsillitis, and malaria. In addition, the resin of *D. buettneri* possesses antimicrobial properties and is used as a disinfectant. The root of *D. klaineana* is used to treat skin conditions [18]. *Dacryodes* species contain several antioxidants including polyphenols that may be beneficial in diseases related to oxidation, such as diabetes [19,20,21,22,23].

### 3.1. Dacryodes klaineana (Pierre) H.J. Lam

*D. klaineana* also known as *Aucoumea klaineana* Pierre., monkey plum or African cherry fruit, is an evergreen tree with a low crown that can grow up to 30 m in height (Figure 1a) [12,24]. When the bark is sliced, *D. klaineana* exudes tear-like drops of gum. Similar to *D. edulis*, the inner bark has a pinkish-brown or reddish-brown color and a turpentine-like scent [25]. In Nigeria, *D. klaineana* blooms in late January or early February, and its fruits ripen between May and June [26]. This species differs from *D. edulis* in that it has fewer leaflets, a smaller inflorescence, and smaller fruits [26]. The wood of *D. klaineana* is utilized in building construction, carts, axe handles, mortars, and is suggested for telegraph poles and railway sleepers [26]. The fruit is in high demand in Nigeria’s eastern region and can be eaten cooked or raw. The pulp is cooked or roasted to produce a type of butter [25]. Additionally, it was claimed that *D. klaineana* provides significant therapeutic and economic benefits for the local population especially because the plant is not poisonous [27]. Boiled roots of *D. klaineana* have traditionally been used to treat skin conditions [7,28], while the pulverized leaves are included in remedies for painful menstruation. In the Ivory Coast, *D. klaineana* leaves are used to treat tachycardia and coughs [25,29,30].

### 3.2. Dacryodes buettneri (Engl.) H.J. Lam

*D. buettneri* (Figure 1b), commonly recognized as “Assia” or “Ozigo”, is an evergreen plant discovered in secondary and deep woods that can reach heights of 50 m. This tree’s bark sheds a transparent, pungent resin upon injury that solidifies to become opaque [18]. The tree is used in the therapy of diarrhea, microbial infections, malaria, constipation, jaundice, and fever, while its resin is used as an astringent and a disinfectant [18,31].

### 3.3. Dacryodes rostrata (Blume) H.J. Lam

*D. rostrata* (Figure 1c) is a native species that is also referred to as “kembayau”, “keramu”, “kedondong kerut” or “keramuuq botatn” in Malaysia, Brunei, the Philippines, and Indonesia [19,32,33,34,35]. Limited scientific information is available on *D. rostata* and this plant has not been extensively investigated for its chemistry and pharmaceutical potential [36]. It is typically found in Southeast Asia in lowland tropical woods that are undeveloped. The tree is a perennial, evergreen plant that, after about ten years, can reach a height of 40–45 m [19,32,33]. A single seed is located in the centre of the ovoid-shaped fruit, and it has a fleshy, tough pulp that is frequently boiled in hot water before ingestion to soften. When the fruit is immature, it is yellowish brown and as it ripens, it turns purplish black. The fruiting period lasts from October to February and local communities harvest the fruits and sell them to marketplaces [15]. *D. rostrata* fruits are additionally stored in salt or soy sauce and served with porridge or rice. Due to their high nutritional value, *D. rostrata* are considered exceptionally useful in preventing malnutrition [19].

### 3.4. Dacryodes peruviana (Loes.) H.J. Lam

*D. peruviana* (Figure 1d) is a species that originates from Ecuador and is commonly known as “copal” in Spanish, and “kunchay” and “wichilla kupall” in Kichwa [37]. This species is found in the Andean and Amazonian regions of Ecuador, between 0 and 2500 m above sea level, with a particularly high concentration in the provinces of Pastaza, Zamora-Chinchipe, Morona-Santiago, and Napo. It is a plant that can grow up to 20–25 m tall with smooth, oval-shaped, greenish-red capsules that are 2–4 cm long and 1.25 cm wide with a pericarp thickness of 0.4 cm [37]. The fruits have three to four valves and one to four seeds that are 1.5 cm long. Birds, monkeys, and humans all consume the fruit [38]. In the Ecuadorian Amazon, the mesocarp and seeds of the fruit are eaten either raw or cooked overheat. The resin of *D. peruviana* is used as an insect repellent and has a pleasant fragrance. It is also burned around houses as it is reported to repel evil spirits and to treat respiratory ailments [37]. The trunk is utilized for constructing houses, cabinets, and carpentry [38]. *Dacryodes peruviana* essential oil has been found to reduce skin inflammation by decreasing inflammatory cytokines, and it is non-toxic to human keratinocytes, making it a promising candidate for treating inflammatory skin conditions [37].
Figure 1Fruits and flowers of *Dacryodes* species. (**a**)—*D. buettneri* [28]; (**b**)—*D. rostrata* [39]; (**c**)—*D. klaineana* [31]; (**d**)—*D. peruviana* [28]; (**e**)—*D. edulis* [40].
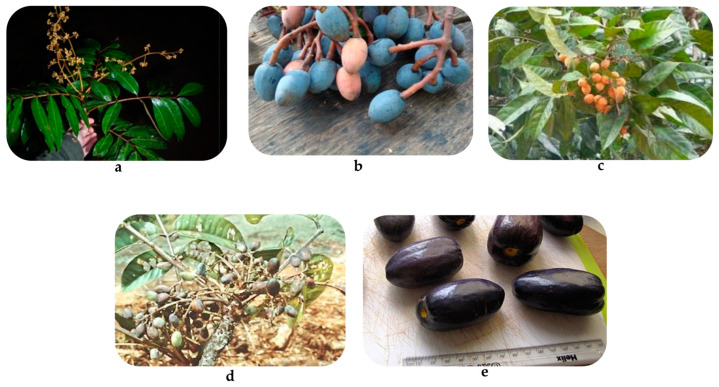



### 3.5. Dacryodes edulis (G. Don) H.J. Lam

The most well-known species in the genus *Dacryodes* is *Dacryodes edulis* (G. Don) H.J Lam (Figure 1e), also referred to as African pear/“Safou”/*Pachylobus edulis* G. Don/*Canarium edule* Engl./Atanga/Bush fruit/Bush butter/Native pear. The Gulf of Guinea and central Africa are home to the attractive and evergreen *D. edulis* tree [21,41,42]. This perennial tree boasts a thick crown and a short trunk, growing up to 18–40 m tall in the wild, but only reaching up to 12 m on farms. It thrives in moist and shaded tropical forests and orchards [21,42], producing large, elongated cylindrical fruits that are initially red or pink when unripe and eventually turn black or blue when ripe, emitting a potent turpentine scent [43].

The tree’s remarkable ability to adapt to varying humidity, soil type, day length, and temperature has made it a widespread crop [42]. Its primary purpose is the cultivation of its delicious fruits, which locals typically boil or simmer in water to soften the pulp before consumption [16]. The tree’s tough, light grey bark is often covered in resin droplets and possesses a pale grey hue [42]. When damaged, the bark releases a transparent resin that eventually solidifies and becomes opaque [21].

*Dacryodes edulis* is valued for its various uses in African traditional medicine [41]. Its fruit is a nutrient-dense source of proteins, vitamins, and lipids and is often consumed alone or with boiled corn. The plant’s stems, roots, bark, and leaves are also used in ethnomedicine to treat a range of illnesses. This is likely due to the high concentration of antioxidants in *D. edulis*, which may explain its effectiveness in treating oxidation-related diseases, such as diabetes, high blood pressure, and cancer [21]. Unfortunately, the *D. edulis* fruit is prone to spoilage due to its quick post-harvest softening, which occurs within 2 to 3 days [44]. In its natural state, the fruit develops and ripens on the tree, without undergoing the softening process. However, once harvested and left at room temperature for a few days, the fruit undergoes rapid softening, which alters its nutritional composition [45].

#### 3.5.1. *Dacryodes edulis* Uses as Food

*Dacryodes edulis* fruit is consumed raw, cooked, or roasted, while the roots, stems, leaves, and bark can be subjected to different processes for various uses. The fruit is typically consumed with bread, corn, rice, cocoyam, or yam due to its high-oil content [46,47]. Because it tends to burst the pulp and squander the contents, intense boiling is avoided. After the skin is removed, the fleshy mesocarp has a buttery texture, it then can be consumed with bread [47]. The pulverized seeds can be used as a seasoning in soups, while the fruit pulp is used to season vegetable soup. Cooking oil can also be made from the fruit, which can replace regular cooking oil [47]. Household ruminant animals like sheep and goats eat the kernel [47].

#### 3.5.2. *Dacryodes edulis* Miscellaneous Uses

The plant is grown around the home in conjunction with cocoa or coffee trees to provide shade [47]. The plant’s trunk can be used to create mortars, axe shafts, and tool handles [48]. Oil is squeezed out of the pulp of the fruit and added to formulations of balm and soap [47]. The dehydrated trunk, twigs, branches, and make excellent fuelwood, and the exudate from the stem is used as fuel to light torches and outdoor lamps [49,50].

#### 3.5.3. Medicinal Purposes

The role of ethnobotanical studies in domesticating, enhancing, and conserving plant-based products cannot be overemphasized. These studies also reveal societal values and perceptions regarding the use and application of bio-resources in communities. However, the transmission of such knowledge between generations is sometimes unreliable, leading to the extinction of valuable information [47]. To prevent the loss of cultural practices and the ability to identify and utilize commercially valuable wild plant species, there is a need to record indigenous ethnobotanical knowledge globally [51].

In Nigeria and neighboring countries, the National Agricultural Research (NARs) and World Agroforestry Centre (ICRAF) are making significant efforts to preserve the knowledge of the *D. edulis* plant, particularly in the southern regions where it is believed to have originated [52,53]. Their objective is to accurately identify, describe, and reproduce the species for successful domestication and enhanced production techniques. These efforts are crucial to ensuring the preservation of valuable ethnobotanical knowledge and sustainable utilization of plant resources.

*Dacryodes edulis* has traditionally been used in a few African countries to nurse fever, diarrhea, wounds, and skin conditions [54]. Anemia and diarrhea are both treated with its bark decoction. The decoction can also be used as a mouthwash or a gargle to treat tonsillitis and maintain good oral hygiene [42]. The resin exudates can be applied alone or combined with regional oils, such as palm oil, to cure various skin diseases. Crushed leaf liquid is also administered in affected areas to treat ectoparasite infection, skin diseases, and other disorders [43,47,48] A lotion, which can be used to smoothen the skin, can be made by adding avocado pulp, palm kernel oil, and spices to the resin exudate [50,55]. Roots that have been cooked with additional herbs are helpful for beriberi and rickets when eaten [47]. For toothache, gum problems, tonsillitis, and earache, bark or leaf decoction is applied [7,29,30]. Elephant grass or other plant components are consumed when boiling leaves or bark of *D. edulis* for antimalarial application [47]. Water made from fermented, ground-up maize is used to boil leaves or bark to create a broth useful in combatting epilepsy and stunted growth in children [47]. After being simmered with regional oils and other ingredients, dried, powdered leaves are administered to ease labor pains or to promote safe delivery. In Cameroon and Nigeria, the decoction of the plant leaves is used in ethnomedicine in the therapy of some disorders, such as earache, toothache, and those in the digestive tract. The stem barks or stem and leaves are used to heal anemia and dysentery [56]. In Southwest Cameroon, the leaves are said to be made into plaster to treat snakebites. [57]. Furthermore, *D. edulis* leaves are often squeezed, and the released juice is used to treat skin diseases such as wounds, rashes, ringworm and scabies, while the stem twigs or stems are used as chewing sticks for oral hygiene [58]. Known in Yoruba as “Ewe pear”, *D. edulis* leaves are used by “Iwo” and “Ibadan” people in South-West Nigeria as a stretch mark remover and to nurse acute malaria and wounds [59]. *Dacryodes edulis* exudates have been reported to be pharmaceutical industry possible feedstock sources [60]. Some of these uses of *Dacryodes* species are summarized in Table 1 and their application in ethnomedicine illustrated in Figure 2.

#### 3.5.4. Spiritual Purposes

The pleasant aroma from burning *D. edulis* components, including the leaves, bark, and root, is assumed to fend off evil spirits by African communities, especially around ill people [47]. Because the plant is viewed as “clean”, worshipers and priests of regional deities frequently use its leaves to wrap sacramental objects and medicinal concoctions. The local idol priests use the leaves as part of their offerings for asking the gods for rainfall and wrapping pieces of bamboo poles and other items for prayers [47].

The lengthy history of using the fruit for sustenance provided most of the evidence supporting its safety, but the safety of other plant components that have been extensively used in traditional medicine is not fully established. Thus, more studies are essential to evaluate the toxicity of *D. edulis* crude extracts [61].

#### 3.5.5. Nutritional Value of *D. edulis*

The edible portion of *D. edulis* is composed of 8.7% dietary fiber, 5% carbohydrates, 4% protein, and 22% oils; its vitamin C content was estimated at 14.11 mg/100 g of crude pulp extract but this composition varies according to some environmental factors [62,63]. The study of the mineral composition of its fruits showed that it is an important source of iron (4.91–8.67 mg/kg), sodium (51.54–108.01 mg/kg), potassium (552.39–646.05 mg/kg), magnesium (23.13–73.18 mg/kg), zinc (130.48–136.18 mg/kg, manganese (25.96–26.86 mg/kg) and calcium (531.31–1337.02 mg/kg), thus indicating the nutritional value of *D. edulis* fruits [64,65,66]. The fatty acid analysis of *D. edulis* pulps showed that they contained almost 49–59% of fatty acid palmitic (41–47%), oleic (20–34%), and linoleic (22–29) acids [63,64,65,66,67]. However, the composition of pulps in some regions of Cameroon was found to fluctuate depending on some environmental factors (climate, soil type, farming practices and fertilizer application) as reported [67], and these factors could also explain the significant variation of percentages of linoleic acid, palmitic acid, oleic acid and stearic acid as several studies revealed in Central and West Africa [63]. Oluwaniyi et al., reported the nutritional value of pulps comparatively to seeds, and as resulted, the pulps were richer in fatty acids (24.60 mg/100 g) than seed (6.48 mg/100 g), but its content in fiber was less than seeds (39.20 and 46.33 mg/100 g, respectively) [68]. The same study also revealed that the pulps contained more vitamin A (612.00 vs. 188.00 mg/g), C (43.56 vs. 12.28 mg/g) and E (54.17 vs. 51.90 mg/g) than seeds, but seeds contained a higher level of vitamin B1 (20.25 vs. 9.29 mg/g), and finally, the mineral content was higher in pulps (3.71 mg/g of Ca vs. 0.43 mg/g in seeds) [68]. This nutritional value has been widely reviewed [12].

Commonly known as “kembayau” fruit, *D. rostrata* could be a good source of energy and minerals for humans, while seeds have antioxidant properties; fruits are submerged in room temperature water for 10–15 min, removed, sprinkled with salt, and devoured; the pulp and seed cotyledons of the fruits are edible [31]. *D. rostrata* peels were found to have a higher percentage of potassium (380.72–1112.00 mg/100 g), followed by pulp (264.45–472.87 mg/100 g) and seeds (196.07–492.09 mg/100 g); peels and seeds were also high in magnesium (52.02 to 62.39 mg/100 g and 70.15 to 235.68 mg/100 g, respectively) [19].

## 4. Plant Distribution

*Dacryodes* species are naturally distributed in tropical America (14 identified species and 22 unidentified species), Central and South Africa (18 species), and Southeast Asia (18 species) (Figure 3) [17]. Equatorial Guinea (six species), Cameroon (ten species), and Gabon (12 species) have the highest diversity of *Dacryodes*. However, only *D. edulis* is widely cultivated among all African species [17]. *Dacryodes edulis* is native to Central Africa’s humid tropical-equatorial zone, mainly in Congo, Democratic Republic of Congo, Gabon, and Cameroon [69]. In Cameroon, it is widely distributed in the West, Adamawa, East, Littoral, and Centre regions, where it is known as “Safou” [70]. In Nigeria, *D. edulis* is primarily found in the southeast and is commonly referred to as “ube” [67]. It is also found in countries stretching from Nigeria to Angola and the eastern part of Uganda, as well as in Sao Tome [42,70,71]. *Dacryodes klaineana* is found in West and Central African humid tropical forests [26], and *D. buettneri* is a common species in the equatorial forest region from Gabon to Equatorial Guinea [31].

*Dacryodes rostrata* is widely found in Malaysia, more precisely in the region of Sabah, Sarawak [34] and on Borneo Island [15,32,36]. It is an Indonesian endemic fruit tree originating in Kalimantan and found in all districts throughout West Kalimantan [72]. In addition, it is also found in Thailand, Indo-China, Sumatra, Sulawesi, the Philippines [31] and Singapore [73]. Three additional species including *Dacryodes rugosa* (Blume) H.J. Lam, *Dacryodes incurvata* (Engl.) H.J. Lam, and *Dacryodes costata* (A.W.Benn.) H. J. Lam are distributed in Singapore, specifically in Nee Soon Swamp Forest [73].

Moreover, nine other species including *Dacryodes ramose* Daly& M. C. Martínez, *Dacryodes froesiana* Daly & M. C. Martínez, *Dacryodes spatulate* Daly & M. C. Martínez, *Dacryodes villosa* Daly & M. C. Martínez, *Dacryodes obovate* Daly & M. C. Martínez, *Dacryodes caparuensis* Daly & M. C. Martínez, *Dacryodes decidua* Daly & M. C. Martínez, *Dacryodes oblongifolia* Daly & M. C. Martínez, and *Dacryodes oblongipetala* Daly & M. C. Martínez were reported to grow in the Guianas and Amazonia. The species *Dacryodes sudyungasensis* Daly & M. C. Martínez was found in Bolivian montane forests on the eastern slopes of the Andes [74]. *Dacryodes uruts-kunchae* Daly& M. C. Martínez & D. A. Neill is distributed in the sub-Andean Cordillera del Cóndor provinces of Mo-rona-Santiago and Zamora-Chinchipe in Ecuador, and in the Andes foothills in the mid-Ro Maraón region of Amazonas and San Martn in Peru [75]. The four species *Dacryodes occidentalis* Cuatrec., *Dacryodes olivifera* Cuatrec., *Dacryodes colombiana* Cuatrec., and *D. peruviana* and are distributed on the Central Cordillera’s eastern slope [76]. Additionally, Martínez-Habibe and Daly reported that *Dacryodes transitionis* M. C. Martínez & Daly, *Dacryodes robusta* M. C. Martínez & Daly, *Dacryodes yaliensis* M. C. Martínez & Daly, *Dacryodes cristalinae* M. C. Martínez & Daly and *Dacryodes canaliculate* M. C. Martínez & Daly, are largely found in Antioquia (Columbia) [77]. The species *Dacryodes talamancensis, D. Santam*. & Aguilar, is found in Central America, more precisely in Costa Rica, Panama, and in the region of Alto Lari in the Caribbean [78]. *Dacryodes excelsa* Vahl. is reported to grow in Tabonuco forest in Puerto Rico [79].

## 5. Commercialization

The well-known “Safou” are produced and commercialized in Cameroon for the local population and internationally. In 1997, production was estimated at 11,000 tons. These fruits are exported from Cameroon to France, Belgium, and the UK [80]. During the production period, the African pear trade produces revenue for neighborhood vendors, putting the sub-region at the center of significant economic activity [81]. The significant economic potential of *D. edulis* can help to alleviate rural population poverty (the estimated value of the commercialization is not reported) [45]. Given its valuable nutritional composition, *D. edulis* pulp could find use in the global food, medicinal, and cosmetic industries [21].

## 6. Conservation of *Dacryodes edulis* Fruits

*Dacryodes edulis* enterprise is affected by its highly perishable nature, which is responsible for significant post-harvest risks as a result of appropriate post-harvest handling. The pulp can be conserved in brine (Containing 2% of NaCl) for 3 months, this solution does not affect the texture and the organoleptic properties of the resulting fruits [82].

## 7. Bioactivity Studies

### 7.1. Antimicrobial Activity

Acetone was used to extract the leaves of *D. edulis*, which were then fractionated into dichloromethane and aqueous fractions. The dichloromethane fraction exhibited a high antibacterial activity with an inhibition zone of 28 mm against *Pseudomonas aeruginosa* (NCIB 950) and *Pseudomonas fluorescens* (NCIB 3756), which was higher than the positive controls, streptomycin (19–27 mm) and ampicillin (0–22 mm) [55]. Previous studies have reported antibacterial activity of the ethyl acetate fraction of *D. edulis* leaves against *Staphylococcus aureus*, *Klebsiella pneumoniae*, and *P. aeruginosa*, with inhibition zones of 29, 28, and 25 mm, respectively [83]. Methanol extraction of *D. edulis* leaves also showed antibacterial activity against various bacteria, including *Bacillus stearothermophilus*, *K. pneumoniae*, *Bacillus polymyxa*, *Citrobacter freundii*, *P. aeruginosa*, *Trueperella pyogenes*, *Enterococcus faecalis*, *Escherichia coli*, *Micrococcus luteus*, *Shigella* sp., and *S. aureus*, with inhibition diameters ranging from 12.0–23.3 mm, compared to 13.0–31.0 and 18.3–29.3.5 mm for streptomycin and ampicillin, respectively [84]. The flavonoid content of *D. edulis* exhibited antimicrobial activity against *Proteus mirabilis* and *Candida albicans* with IC_50_ values of 5 and 10 mg/mL, respectively [85]. Other studies summarized the antimicrobial activity of *D. edulis* against several microbes, including *Bacillus subtilis*, *Escherichia coli*, *Klebsiella aerogenes*, *S. aureus*, *Salmonella typhi*, *P. aeruginosa*, *S. enteric*, and *Proteus mirabilis* [13]. Additionally, the essential oil extract of *D. peruviana* fruits displayed antibacterial activity with a MIC value of 625 µg/mL against *S. aureus*, compared to 15.62 µg/mL for Tetracycline [38].

### 7.2. Anticancer Activity

The extract from *D. edulis* leaves has demonstrated effective antiproliferative properties against ovarian cells in mammals. This extract was able to reverse the effects of oestradiol by reducing the weight of the uterus and the height of the epithelium of the uterus and vagina [86]. There have been numerous reports indicating that estrogen receptors are linked to approximately 80% of breast cancers [87]. Mvondo et al., showed that the aqueous extract of *D. edulis* leaves reduced the tumor volume in female rats with breast cancer induced by 7,12-dimethylbenz[α]anthracene (DMBA; 50 mg/kg BW) [88]. In vitro, the cytotoxic potential of *D. rostrata* peels on breast cancer cell lines was determined using the MTT method. The results showed that the peel water extract had a moderate cytotoxic effect on T-47D breast cancer cells (with IC_50_ values of 143.02 ppm and 322.55 ppm, respectively) [71].

### 7.3. Antidiabetic Activity

The fruit extract of *D. edulis* has been shown to be effective in managing diabetes in Alloxan-induced diabetic rats. Sub-toxic doses of the hexane extract (400 to 1600 mg/kg) administered orally resulted in a significant decrease in overall cholesterol, triglycerides, hyperglycaemia, ALT, LDL-C, and ALP levels [53]. Additionally, the aqueous-methanol fraction of the *D. edulis* leaf extract demonstrated significant α-amylase and α-glucosidase inhibitory activity in vitro. In streptozotocin-induced diabetic Albino Wistar rats, the crude extract, H_2_O-MeOH, and EtOAc fractions significantly decreased fasting blood glucose levels from 463 to 174 mg/dL, 427.8 to 151 mg/dL, and 460 to 120.5 mg/dL, respectively, revealing the anti-hypoglycemic potential of *D. edulis* leaves, respectively, in streptozotocin-induced diabetic Albino Wistar rat model, revealing the anti-hypoglycaemia potential of *D. edulis* leaves [89]. Furthermore, Erukainure et al., reported that the n-butanol fraction of *D. edulis* leaves had a significant effect on type 2 diabetes induced by fructose-streptozotocin in rats compared to the standard drug metformin. This illustrated the ability of this fraction to decrease the level of glucose in the blood, improve β-cell dysfunction, serum insulin, inhibit insulin resistance, and improve weight and morphology of the pancreas [90].

### 7.4. Antioxidant and Free Radicals Scavenging Activity

*Dacryodes edulis* methanol extract of leaves was prepared, then H_2_O-MeOH, EtOAc, and hexane fractions were prepared for antioxidant assays. As a result, H_2_O-MeOH and EtOAc fractions exhibited pronounced radical scavenging and reductive activity in vitro. Additionally, further in vivo tests showed the capacity of the crude extract and the resulting fractions to reduce the oxidative stress burden in diabetic rats [89]. Nguefack reported that the *D. edulis* ethanol extract promotes the restorative activity in oxidative stress CCl_4_-induced rats and in tissue (blood and liver) damage [91]. *Dacryodes edulis* aqueous extract of seeds was assayed for its antioxidant potential by Ogunmoyole et al., and it was found that the extract obtained at 100 °C had better activity than the extract obtained at room temperature with 65% and 40% of free DPPH free radicals scavenged, respectively. The similar results were obtained with FRAP method where the boiled extract reduced more than 75% of Fe^3+^ comparatively to 35% of unboiled extract [57]. These results therefore indicated the high potential of decoction traditionally used by tradipractitioners. The methanol extract of *D. edulis* seed was evaluated for its antioxidant activity, scavenging, 37% of free radicals relative to vitamin C (100%) and *β*-tocopherol (98%). It also exhibited significant concentration-dependent AChE inhibitory activity [92]. Furthermore, Uhunmwangho and Omoregie assessed the antioxidant properties of the fruit of *D. edulis* during the maturation from 4 weeks after anthesis to mature fruits (20 weeks) following three methods namely: H_2_O_2_, 2,2-diphenyl-1-picrylhydrazyl (DPPH) radical-, and malondialdehyde (MDA) assays. This resulted in the CHCl_3_ /methanol (1:2) extract scavenging 45.47, 45.1 and 18.3% of radicals in the three methods, respectively, and the activities were higher with mature fruit [93]. Jolayemi et al., also reported the antioxidant potential of pulp extract [61]. The flavonoid fruit extract was reported to have higher radical scavenging activity (IC_50_ = 0.223 mg/mL) compared to vitamin C (IC_50_ value = 0.299 mg/mL), quercetin (IC_50_ value = 0.334 mg/mL) and butylate hydroxyanisol (IC_50_ value = 0.318 mg/mL) [85].

Based on the previous IC_50_ values obtained, it is evident that the antioxidant activity of *D. edulis* is directly linked to its flavonoid content. Additionally, the flavonoid content in *D. edulis* fruits can be explored for potential therapeutic applications. The ex vivo anti-oxidative activity of *D. edulis* extract was tested on pancreas and liver tissues collected from male albino rats. These organs were first injured with FeSO_4_, and untreated injured hepatic tissue exhibited an increase in glucose-6-phosphatase activity, indicating an increase in glycogenolysis and gluconeogenesis. The important (*p* < 0.05) reduction in this activity, due to the use of the aforementioned extract, highlights their antihyperglycemic potential with an increase in glycogenesis [60]. The in vitro antioxidant activity revealed that the *D. edulis* ethanol extract was more productive (IC_50_ value of 1.83 µg/mL) than the aqueous extract (44.57 µg/mL) l-ascorbic acid (8.27 µg/mL) using the DPPH method; even though the activity of the ethyl acetate fraction was the smallest, this fraction was more effective in vivo on the activities of several enzymes such as GSH, SOD, and Alpha glucosidase [60]. Overall, the effectiveness of *D. edulis* as an antioxidant depends on various factors, including the solvent used, extraction temperature, and assay method (in vivo or in vitro). Nevertheless, these results suggest that therapeutic investigations should prioritize the flavonoid content, the ethanolic and ethyl acetate extracts, high-temperature extraction, and in vivo assays. For traditional purposes, people should use an alcoholic solvent like palm wine to enhance its activity in treating diseases associated with free radicals.

The defatted 50% ethanol extracts of each part of *D. rostrata* were assayed for their antioxidant activities. The seeds exhibited higher activities than peels followed by pulps, and this was following the phenolic contents of these parts [62]. The ethanol extract of seeds has also been found to contain higher antioxidant properties in comparison with that of pulp and peel extract [19]. Furthermore, the EtOAc fraction from the hydroethanolic (50%) extracts exhibited the highest activities among all the tested fractions (H_2_O, *n*-BuOH, and EtOAc), yielding similar results for each part of the fruits. A recent report on the antioxidant potential of H_2_O-EtOH (50%) extract of the peels of *D. rostrata* showed that they exhibited an activity with IC_50_ values of 59.27 ppm and 121.7 ppm in the 2,2-diphenyl-1-picrylhydrazyl assay, which is classified as strong and moderate activity, respectively [71]. Thavamoney et al., reported the antioxidant potential of the fractions resulting from the fractionation of H_2_O-EtOH (50%) extract of these fruits, stating that the crude extract of each part had higher activity than any of the fractions [30]. These results on *D. rostrata* support the findings previously suggested for *D. edulis* concerning the use of alcoholic solvents.

### 7.5. Effect on Hepatocytes

*Dacryodes edulis* seed’s ethanol extract was reported to have an hepatoprotective effect. Albino rats of the Wistar strain were treated with the seed extract for two weeks and then CCl_4_ on the fourteenth day, showing no liver damage, whereas rats that were only subjected to CCl_4_ showed liver damage [94]. Similarly, the same authors assessed the hepatoprotective activity of the seeds aqueous extract [95]. The liver section of male Wistar rats with CCl_4_-induced hepatotoxicity, treated with ethanol extract of *D. edulis* seeds, highlighted the potential of this extract to restore hepatocyte functions [96]. Furthermore, the hepatoprotective activity of the fruits of *D. edulis* was evaluated during their maturation from four weeks after anthesis to mature fruits on male Wister rats. The CHCl_3_/methanol (1:2) oil extract used to feed rats showed activity in increasing order from immature fruits to mature ones [93], thus indicating that the mature fruits are more hepatoprotective.

### 7.6. Toxicity

The toxicity of ethyl acetate and the aqueous fractions of *D. edulis* leaf was previously evaluated on male Wistar rats. After 24 h, no damage was observed on their livers and kidneys, thereby indicating the safety of the leaf extract [83]. Tene et al., 2016 also reported that any part of this plant can be eaten without risk of toxicity [13]. In a recent study, *D. edulis* methanolic leaf extract decreased erythrocytic indices at high doses indicating anemic potentials; however, liver and kidney function indices were similar to control levels [2].

To sum up, these findings on the biological properties of *Dacryodes* species provide evidence for their traditional use in treating various ailments. Additionally, other species in the genus should be investigated for their potential therapeutic benefits. Most of the reported assays have employed outdated antioxidant test methods, and hence, modern techniques are required to better comprehend the biological activities of these species. Furthermore, future in vitro and in vivo studies should focus on the activities of isolated flavonoids, as well as phytochemicals belonging to other classes of compounds, or to be isolated. Table 2 summarizes the reported biological activities above discussed.

## 8. Phytochemistry

Since plant chemical constituents are physiologically active, research into these substances is crucial for finding new drugs and creating novel therapeutics for medical disorders [97]. Over the years, the medicinal properties of *D. edulis* have been consumed in rural communities throughout the tropical regions. Different species of the *Dacryodes* genus have bioactive compounds, such as saponins, terpenoids, flavonoids, tannins, reducing sugars, and alkaloids, justifying their uses in ethnomedicine. The tree was screened quantitatively and qualitatively to identify the phytochemical elements accountable for the ethnomedicinal properties [42] (Figure 4 and Table 3). The compounds **1**–**14** identified as α-Thujene (**1**), α−Pinene (**2**), Camphene (**3**), Sabinene (**4**), β-Pinene (**5**), α−Phellandrene (**6**), α-Terpinene (**7**), *p*- Cymene (**8**), 1,8-Cineole (**9**), γ−Terpinene (**10**), *cis* Sabinene hydrate (**11**), *trans* Sabinene hydrate (**12**), Terpinen-4-ol (**13**) and α−Terpineol (**14**) were reported from the essential oil of the *D. edulis* resin [21]. The compounds **15**–**19** i.e., Methyl gallate (**15**) [98], Ellagic acid (**16**) [99,100], Quercetin (**17**) [100], Quercetin 3-*O*-α-l-rhamnoside (**18**) [100] and Catechol (**19**) [100], were identified from the fruits of *D. edulis* [100], while Afzelin (**20**) and Sitosterol-3-*O*-β-d-glucopyranoside sterol (**21**) were from the stem bark of *D. edulis* [41]. Gallic acid (**22**) was isolated from the seeds of *D. edulis* [101,102] and the compounds **23**–**25** i.e., Vanillic acid (**23**) [103], Vanillin (**24**) [104], (-)-epicatechin (**25**) [23,90] were identified from the leaves of *D. edulis*. From the same species, the following compounds: Kaur-15-ene (**26**) [61], 9-(4-methoxyphenyl) xanthene (**27**) [105], Xanthone (**28**) [106], Octadecanoic acid (**29**) [107], Phytol acetate (**30**) [108], Ethyl-15-methylheptadecanoate (**31**) [61], *trans* Phytol (**32**) [108], Ascorbic acid-2,6-dihexadecanoate (**33**) [61], Urs-12-ene-3-ol acetate (**34**) [61], 2,3,23-trihydroxyolean-12-en-28-oic acid methylester (**35**) [61], and Sitosterol (**36**) [109] were identified from the leaves and reported to exhibit good binding affinity towards α-glucosidase [60] (Table 3). Finally, β-amyrin (**37**) was isolated from the stem bark of *D. edulis* [110]. Table 3 and Figure 4 show a summary of some of the compounds that have been identified from *D. edulis*, as well as their biological activities.

## 9. Summary

This review covers the uses of various *Dacryodes* species, including traditional, domestic, and other miscellaneous uses. Out of the 70 species of *Dacryodes*, most have limited scientific information available. Therefore, this review focuses more on *D. edulis* and *D. rostata*, from which only a few compounds have been successfully isolated and identified as belonging to the monoterpene, triterpene and phenolics class.

The findings on the biological activities of *Dacryodes* species support their traditional use in managing various ailments. Additionally, other species should be explored for their potential. Further research should investigate the activities of isolated flavonoids from these plants and conduct in vivo assays.

## 10. Conclusions

The purpose of this review is to discuss the ethnobotanical uses, distribution, pharmacology, and phytochemistry of *Dacryodes* species. Among the various species of *Dacryodes*, two, namely *D. edulis* and *D. rostrata*, are well-known worldwide for their edible fruits and medicinal properties. *Dacryodes edulis* is primarily distributed in Central Africa and has been found to contain potential phytomedicines for treating diabetes, cancer, microbial infections, tonsillitis, and oxidative stress-related diseases based on in vivo studies. This is in line with its use in ethnomedicine for treating a range of diseases. On the other hand, *D. rostrata* is predominantly found in tropical Southeast Asia, including Indonesia, Malaysia, the Philippines, and Thailand. Reports confirm the antioxidant activities of different parts of its fruits, including the pulp, peels, and seeds. Compounds identified from *D. edulis* include sabinene, trans-hydrate sabinene, terpinene-4-ol, methyl gallate, ellagic acid, and cis-hydrate sabinene. Meanwhile, vanillic acid, vanillin, and (−)-epicatechin are among the compounds identified from *D. rostrata*, which has only been investigated for its antioxidant activities. These findings highlight the potential of the *Dacryodes* genus as a source of pharmacologically and medicinally important chemicals, making it a promising alternative treatment for various ailments and a starting material for developing new pharmaceutical products.

The biological activities discovered in *Dacryodes* species support their use in traditional medicine by rural communities. However, the pharmacological preparations from trees of this genus have unclear prescriptions and mechanisms of action, and their efficacy, as well as related side effects, are not yet fully recorded or understood. Therefore, extensive further research is needed to establish this missing knowledge. Although there has been in vitro research on the biological properties of *Dacryodes* species, there is limited information on the biological properties of their chemical constituents. More research is required on phytochemistry, pharmacology, and toxicology, as well as in vivo experiments using isolated chemical compounds and crude extracts from the species. Molecular docking investigation, metabolomics, and high-throughput screening could be used to gain further insights into the phytochemicals produced by *Dacryodes* trees and their associated biological effects.

## Figures and Tables

**Figure 2 pharmaceuticals-16-00775-f002:**
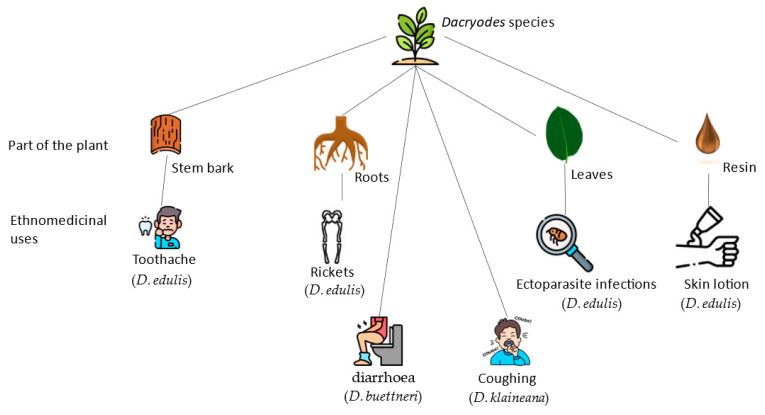
Illustration of *Dacryodes* species and their application in ethnomedicine.

**Figure 3 pharmaceuticals-16-00775-f003:**
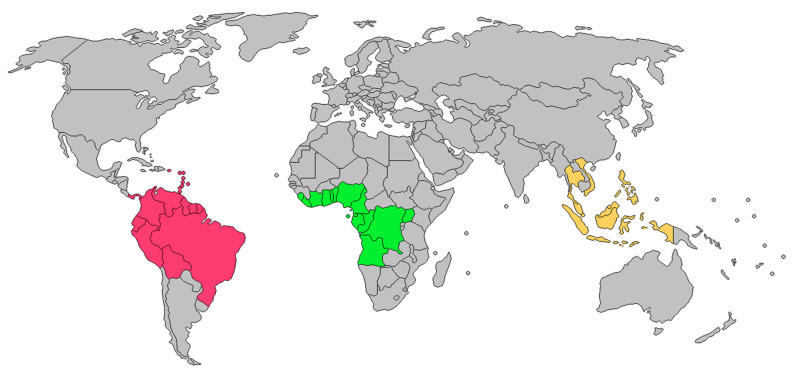
Countries with reports of *Dacryodes* species. Pink = South American species like as *D. excelsa* and *D. peruviana*, green = African species such as *D. edulis* and *D. klaineana*, yellow = Southeast Asian species such as *D. rostrata* and *D. costata* [15,17,26,31,32,34,36,42,67,69,70,71,72,73,74,75,76,77,78,79].

**Figure 4 pharmaceuticals-16-00775-f004:**
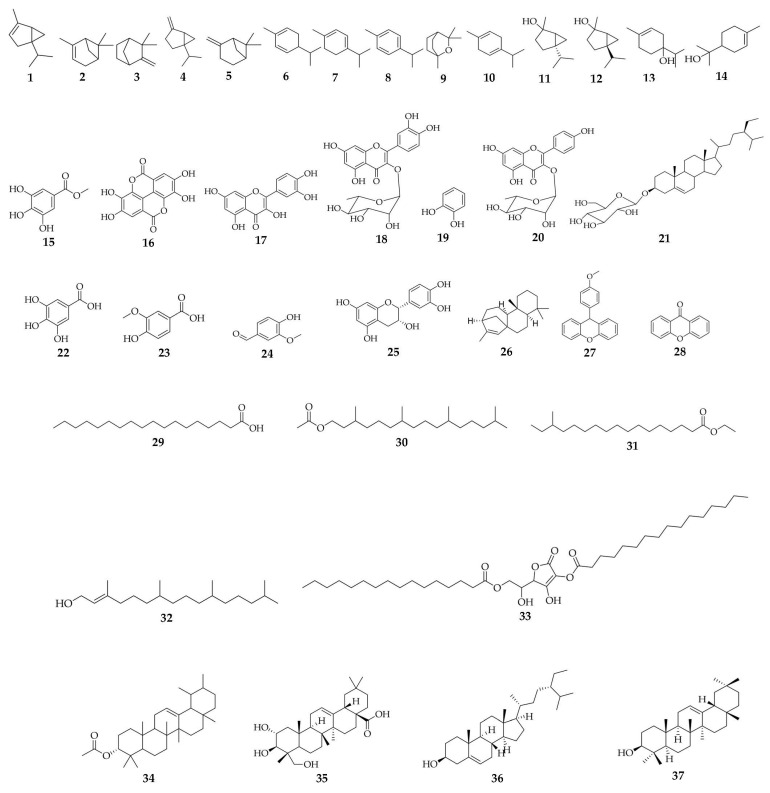
Compounds identified from *D. edulis*.

**Table 1 pharmaceuticals-16-00775-t001:** Summation of ethnomedicinal studies reported on species of the *Dacryodes* genus.

Species	Plant Part	Ethnomedicinal Use	References
*D. edulis*	Stem bark	For toothache, gum problems, gargling, mouthwash, tonsillitis, and earache, bark or leaf decoction is applied.	[7,29,30]
	Roots	Cooked with additional herbs and eaten to combat beriberi and rickets	[47]
	Leaves	Administered For skin diseases, ectoparasite infections, and disorders.	[43,48,60]
	Resin	Exudes is applied to cure skin diseases (ringworms, craw-craw and wounds).	[43,48,60]
*D. klaineana*	Roots	Treatment of skin conditions by consuming them boiled.	[7,28]
	Leaves	Remedy for painful menstruation; tachycardia and cough.	[25,29,30]
*D. buettneri*	Not reported	Used for the treatment of diarrhea, microbiological infections, malaria, constipation, jaundice, and fever, its resin is used as an astringent and a disinfectant.	[18,31]

**Table 2 pharmaceuticals-16-00775-t002:** Biological activities of the plants of the genus *Dacryodes*.

Plants (Organs)	Biological Activities	Mode of Assessment	References
*D. edulis* (Fruits)	Antidiabetic	In vivo	[53]
*D. edulis* (Fruits)	Antioxidant	In vitro	[93]
*D. edulis* (Fruits)	Antihyperglycemic	Ex vivo	[60]
*D. edulis* (Leaves)	Antibacterial	In vitro	[55,83,84]
*D. edulis* (Leaves)	Antiproliferative	In vivo	[86]
*D. edulis* (Leaves)	Antitumoral	In vivo	[88]
*D. edulis* (Leaves)	α-amylase and α-glucosidase inhibition	In vitro	[89]
*D. edulis* (Leaves)	Anti-hypoglycaemia	In vivo	[89]
*D. edulis* (Leaves)	Antidiabetic	In vivo	[90]
*D. edulis* (Leaves)	Antioxidant	In vitro	[89,91]
*D. edulis* (Pulps)	Antioxidant	In vitro	[61]
*D. edulis* (Seeds)	Radical scavenging	In vitro	[57]
*D. edulis* (Seeds)	Antioxidant	In vitro	[91]
*D. edulis* (Seeds)	Hepatoprotective	In vivo	[94,96]
*D. peruviana* (Fruits)	Antibacterial	In vitro	[38]
*D. rostrata* (Seeds, peels, Pulps)	Antioxidant	In vitro	[19,62]
*D. rostrata* (Peels)	Antioxidant	In vitro	[71]
*D. rostrata* (Fruits)	Antioxidant	In vitro	[30]
*D. rostrata* (Peels)	Cytotoxic	In vitro	[71]

**Table 3 pharmaceuticals-16-00775-t003:** Compounds identified from *D. edulis* and their biological activities.

Compound Name	Molecular Formula	Plant Part	Biological Activities	References
α-Thujene (**1**)	C_10_H_16_	Resin	Antioxidant, antimalarial, antibacterial, antimicrobial, and herbicidal activities	[21]
α−Pinene (**2**)	C_10_H_16_	Resin	Antimicrobial, apoptotic, antimetastatic, antibiotic, and antiinflammatory properties.	[21]
Camphene (**3**)	C_10_H_16_	Resin	Antifungal potential against certain fungi when combined with sage oil and has the potential to act as an antioxidant when combined with vitamin C and citrus oils.	[21]
Sabinene (**4**)	C_10_H_16_	Resin	Antifungal and antiinflammatory activities.	[21]
β-Pinene (**5**)	C_10_H_16_	Resin	Fungicidal agent, antiviral and antimicrobial agents.	[21]
α−Phellandrene (**6**)	C_10_H_16_	Resin	Reduce pain sensitivity and increase energy levels. Contains potential anticancer properties.	[21]
α-Terpinene (**7**)	C_10_H_16_	Resin	Antioxidant, anticancer, anticonvulsant, antiulcer, antihypertensive, and antinociceptive	[21]
*p*- Cymene (**8**)	C_10_H_14_	Resin	Used to prevent coughs and eliminate phlegm.	[21]
1,8-Cineole (**9**)	C_10_H_18_O	Resin	The respiratory tract is mucolytic and spasmolytic. It has been shown to be beneficial in the treatment of inflammatory airway diseases such as asthma and chronic obstructive pulmonary disease (COPD).	[21]
γ−Terpinene (**10**)	C_10_H_16_	Resin	Antioxidant	[21]
*cis* Sabinene hydrate (**11**)	C_10_H_18_O	Resin	Antiinflammatory and antifungal properties.	[21]
*trans* Sabinene hydrate (**12**)	C_10_H_18_O	Resin	Antiinflammatory and antifungal properties.	[21]
Terpinen-4-ol (**13**)	C_10_H_18_O	Resin	Antiinflammatory, anticancer, and antioxidant agents	[21]
α−Terpineol (**14**)	C_10_H_18_O	Resin	Antioxidant, anticancer, anticonvulsant, antiulcer, antihypertensive, antinociceptive activities	[21]
Methyl gallate (**15**)	C_8_H_8_O_5_	Fruits	Antioxidant activity	[98,100]
Ellagic acid (**16**)	C_14_H_6_O_8_	Fruits	Cancer prevention, antiviral and antibacterial activities	[99,100]
Quercetin (**17**)	C_15_H_10_O_7_	Fruits	Antioxidant and anticancer properties	[41,100]
Quercetin 3-*O*-α-l-rhamnoside (**18**)	C_21_H_20_O_11_	Fruits	Antiinflammatory and antioxidant effects that might assist to control blood sugar, kill cancer cells, lessen swelling, and help to prevent heart disease.	[100]
Catechol (**19**)	C_6_H_6_O_2_	Fruits	Antioxidant, Antiinflammatory, and antimicrobial agent	[100]
Afzelin (**20**)	C_21_H_20_O_10_	Stem bark	Protect human keratinocytes from the harmful effects of UV irradiation via its biological properties (Antioxidant, Antiinflammatory and DNA-protective) and also act as a UV absorber.	[41,98]
Sitosterol-3-*O*-β-d-glucopyranoside sterol (**21**)	C_35_H_60_O_6_	Stem bark	Treat endotoxemia and inflammation accompanied by the overproduction of nitric oxide.	[41,98]
Gallic acid/3, 4, 5-trihydroxybenzoic acid (**22**)	C_7_H_6_O_5_	Seeds	Antibacterial and antifungal properties	[101,102]
Vanillic acid (**23**)	C_8_H_8_O_4_	Leaves	Antioxidant, hepatoprotective, cardioprotective, and antiapoptotic activities	[84,90,103]
Vanillin (**24**)	C_8_H_8_O_3_	Leaves	Antioxidant and antimicrobial properties	[90,103,104]
(−)-epicatechin (**25**)	C_15_H_14_O_6_	Leaves	Insulinogenic	[23,90]
Kaur-15-ene (**26**)	C_20_H_32_	Leaves	Not reported	[61]
9-(4-methoxyphenyl) xanthene (**27**)	C_20_H_16_O_3_	Leaves	Xanthene derivatives display neuroprotector, antitumor, antimicrobial properties	[61,105]
Xanthone (**28**)	C_13_H_8_O_2_	Leaves	Antitumor	[61,106]
Octadecanoic acid (**29**)	C_18_H_36_O_2_	Leaves	Antiinflammatory	[61,107]
Phytol acetate (**30**)	C_22_H_42_O_2_	Leaves	Phyton derivatives display cytotoxic, antianxiety, antioxidant, metabolism-modulating, autophagy- and apoptosis-inducing, antinociceptive, antimicrobial effects, antiinflammatory, immune-modulating.	[61,108]
Ethyl 15-methylheptadecanoate (**31**)	C_20_H_40_O_2_	Leaves	Not reported	[61]
*trans* Phytol (**32**)	C_20_H_40_O	Leaves	Phytol derivatives display cytotoxic, metabolism-modulating antianxiety, antioxidant, antinociceptive, immune-modulating antiinflammatory, autophagy- and apoptosis-inducing, and antimicrobial effects	[61,108]
Ascorbic acid 2,6- dihexadecanoate (**33**)	C_38_H_68_O_8_	Leaves	Not reported	[61]
Urs-12-ene-3-ol acetate (**34**)	C_32_H_52_O_2_	Leaves	Not reported	[61]
2,3,23-trihydroxyolean-12-en-28-oic acid methyl ester (**35**)	C_30_H_48_O_5_	Leaves	Not reported	[61]
Sitosterol (**36**)	C_29_H_50_O	Leaves	Antinociceptive, analgesic, immunomodulatory, antimicrobial, anticancer, anxiolytic & sedative effects, antioxidant, and antidiabetic activity, antiinflammatory, hypolipidemic, respiratory disease protection, wound healing effect, hepatoprotective.	[61,109]
*β*-amyrin (**37**)	C_30_H_50_O	Stem bark	Not reported	[110]

## Data Availability

No new data were created or analyzed in this study. Data sharing is not applicable to this article.

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
