# Peer review of "The Genus Dacryodes Vahl.: Ethnobotany, Phytochemistry and Biological Activities"

_pharmaceuticals, 2023, doi:10.3390/ph16050775_

Round 1

Reviewer 1 Report

The manuscript entitle "The genus Dacryodes: ethnobotany, phytochemistry and biological activities" is interesting and useful for both medical and biological fields. I have the following comments to improve its quality:

1.     The authors mentioned in the Abstract part, “Members of this genus are known as ethnopharmacological remedies for a variety of ailments such as tonsillitis” kindly add more ethnopharmacological uses not only tonsillitis.

2.     The authors mentioned in the Abstract part “the data reported were systematically sourced, reviewed, discussed, and analyzed from scientific electronic databases such as Scifinder, Scopus, Pubmed, Springer Link, ResearchGate, Ethnobotany Research and Applications, Google Scholar, and ScienceDirect.” Add the period of data collection you used for example 1970-2023.

3.     The authors mentioned in the Abstract part “D. edulis isolates contain phytochemical compounds such as triterpenoids and monoterpenoids with anti-inflammatory properties.” Correct this sentence as triterpenoids and monoterpenoids, not compounds but phytochemical groups belonging to the terpenoids (isoprenoid) class.

4.     The authors mentioned in the Abstract part “Nevertheless, the therapeutic potential of most of the plants in the genus have been underexplored with regards to medicinal chemistry and pharmacology exploits” correct this sentence to “Nevertheless, the therapeutic potential of most of the plants in the genus have been underexploited with regards to phytochemistry and pharmacology.”

5.     Delete reference one as you gave in this sentence statistical data about the importance of medicinal plants while the reference is not updated (2002).

6.     Also, replace out-of-date references with updated references as often as possible to keep your review up-to-date.

7.     In the 2. Literature Survey Databases add the period of data collection.

8.     Correct all the Latin scientific names to be the first letters of the taxonomist not Italic as Dacryodes buettneri (Engl.) H.J. Lamin replace with Dacryodes buettneri (Engl.) H.J. Lam

9.     The whole manuscript needs major grammar, typo and editing corrections by a native speaker specialist in biological and biomedical sciences.

Author Response

Dear Reviewer,

Please see attachement!

Reviewer 2 Report

Pharmaceuticals journal

Review: “The genus Dacryodes: ethnobotany, phytochemistry and bio-logical activities”

1) The introduction section contains general topic about traditional medicine and folk herbal medicine, and there is no relationship to genus Dacryodes that is the subject of the study.

2) It seems that the article only talks about one plant of Dacryodes genus; Dacryodes edulis. There are not many species in this genus that can include an a review topic on this genus. Therefore, I suggest, it is better to write a review only about Dacryodes edulis.

3) There are many mistakes in the formulation and form of scientific writing.

Author Response

Dear Reviewer,

Please see attachement!

Reviewer 3 Report

Dear Authors,

i have the following comments to your work:

1) please, work on the introduction - it is too general and too lengthy. also, it contains many grammatical errors, repetitions of verbs and spelling errors. please, focus on the selected species and on the gender and be more to the point in this section of the manuscript

2) please, rearrange section 3. at the beginning you present a characteristic of the genus and of the selected species, later you mention pharmacological properties and you come back again to the definition of the plant's name and location. please, first describe the botanical characeteristics and later discuss ethnopharmacology

3) section 3.5.5, section 4 and throughout the manuscript -  please use italics for Latin names

4) please, add a table that colelcts all biological activities described in the text; 

5) create a simple figure that shows the major pharmacological activities - the directions for the application of this plant in therapy

6)please, add an additional column to tbale 2 that presents the molecular formulas of the indicated compounds. it will allow an easier screening for active components by other researchers

7) please, rewrite the section 'summary' - it contains may errors. also, the sentence : The similarity of the compounds....  should be removed. please, mention here the major groups of secondary metabolites identified in the plant

Author Response

Dear Reviewer,

Please see attachement!

Round 2

Reviewer 1 Report

The authors conducted all the required corrections, thank you

Author Response

Dear Reviewer,

Thank you very much!

Reviewer 2 Report

1) The revised manuscript's readability is not clear. Add a revision copy in which the changes are marked in color.

2) In table 2; order the first column according to the species.

3) “Figure 34. Compounds isolated or identified from D. edulis and D. rostrata”. This is not clear. Make this into two figures, one for compounds of D. edulis and the other for D. rostrata.

Author Response

Dear Reviewer,

Please see below:

Reviewers Comments to Author:

Reviewer #2:

Concern 1: The revised manuscript's readability is not clear. Add a revision copy in which the changes are marked in color.

Response: We appreciate you bringing this matter to our attention. Based on your suggestions, we have formatted the changes made to the manuscript and the revised texts are now in red color.

Concern 2: In table 2; order the first column according to the species.

Response: Thank you for your insightful comment. We have re-ordered the column according to the species. To see the changes, please see page 13, Table 2.

Concern 3: “Figure 4. Compounds isolated or identified from D. edulis and D. rostrata”. This is not clear. Make this into two figures, one for compounds of D. edulis and the other for D. rostrata.

Response: Thank you for your suggestion. We have made the necessary changes as per your suggestions. Please see page 14, Figure 4.

Best regards,

Xavier

Reviewer 3 Report

Dear Authors

thank you for the corrections that were introduced to the text and for addressing all my comments.

Author Response

Dear Reviewer,

Thank you very much!

Round 3

Reviewer 2 Report

- Fotos in figure 1 is not clear.

- in figure 2, it is better if the name of the species of Dacryodes is written under each ethnomedicinal use.

Author Response

Dear Reviewer,

Reviewers Comments to Author:

Reviewer #2:

Concern 1: Fotos in figure 1 is not clear.

Response: Thank you very much for your constructive comments to improve our paper. The quality of Figure 1 has been substantially improved by replacing unclear photos with high resolution ones.

Concern 2: In figure 2, it is better if the name of the species of Dacryodes is written under each ethnomedicinal use.

Response: Thank you for your insightful comment. We have incorporated the name of species of Dacryodes under each ethnomedicinal use as suggested in Figure 2.